# Kidney damage and associated risk factors in the rural Eastern Cape, South Africa: A cross-sectional study

**Ernesto Rosales Gonzalez** [1]*, **Parimalanie Yogeswaran**[1], **Jimmy Chandia**[1], **Guillermo Alfredo Pulido Estrada**[2], **Oladele Vincent Adeniyi**[3]

**1** Department of Family Medicine and Rural Health, Faculty of Medicine and Health Sciences, Mthatha, South Africa, **2** Department of Public Health, Faculty of Medicine and Health Sciences, Mthatha, South Africa, **3** Department of Family Medicine and Rural Health, Faculty of Medicine and Health Sciences, East London, South Africa

* ernestorosalesgonzalez@gmail.com.

**Data Availability Statement:** The restrictions on sharing the participant's data are ethical and imposed by the Walter Sisulu University EC Research and Ethics Committee (HREC). If you

## Abstract

### Background

The colliding epidemic of infectious and non-communicable diseases in South Africa could potentially increase the prevalence of kidney disease in the country. This study determines the prevalence of kidney damage and known risk factors in a rural community of the Eastern Cape province, South Africa.

### Methods

This observational cross-sectional study was conducted in the outpatient department of the Mbekweni Community Health Centre in the Eastern Cape between May and July 2022. Relevant data on demography, medical history, anthropometry and blood pressure were obtained. The glomerular filtration rate was estimated using the Chronic Kidney Disease Epidemiology Collaboration Creatinine (CKD-EPI$_{Creatinine}$) equation and the re-expressed four-variable Modification of Diet in Renal Disease (MDRD) equation, without any adjustment for black ethnicity. Prevalence of kidney damage was defined as the proportion of individuals with low eGFR (<60mL/min per 1.73m$^2$). The presence of proteins in the spot urine samples was determined with the use of test strips. We used the logistic regression model analysis to identify the independent risk factors for significant kidney damage.

### Results

The mean (±standard deviation) age of the 389 participants was 52.3 (± 17.5) years, with 69.9% female. The prevalence of significant kidney damage was 17.2% (n = 67), as estimated by the CKD-EPI$_{Creatinine,}$ with a slight difference by the MDRD equation (n = 69; 17.7%), while the prevalence of proteinuria was 7.2%. Older age was identified as a significant risk factor for CKD, with an odds ratio (OR) = 1.08 (95% confidence interval [CI]: 1.06–1.1, p < 0.001). Hypertension was strongly associated with proteinuria (OR = 4.17, 95% CI 1.67–10.4, p<0.001).

have any inquiries about our ethical research guidelines, please feel free to contact Professor E. Ndevia, head of the Walter Sisulu University EC Research and Ethics Committee (HREC), at endevia@wsu.ac.za. The ethics committee also makes data from the study available upon request. Please contact Professor E. Ndevia. His contact information is endevia@wsu.ac.za.

**Funding:** The research project was supported by the Discovery Foundation awards The Award to PY (Reference: 035996). https://www.discovery.co.za/corporate/discovery-foundation-awards The Discovery Foundation had no role in study design, data collection and analysis, decision to publish, or preparation of the manuscript.

**Competing interests:** The authors have declared that no competing interests exist.

## Conclusions

This study found a high prevalence of kidney damage (17.2%) and proteinuria (7.97%) in this rural community, largely attributed to advanced age and hypertension, respectively. Early detection of proteinuria and decreased renal function at community health centres should trigger a referral to a higher level of care for further management of patients.

## Introduction

Chronic kidney disease (CKD) is a growing problem in developing countries [1, 2]. In most of sub-Saharan Africa, most patients with CKD die because of a lack of adequate treatment and renal replacement therapy (RRT) [3]. RRT is very expensive, making it unaffordable to people in low- and middle-income countries. Evidence suggests an increase in the demand for RRT in South Africa, from 70 per million of the population in 1994 to 190 per million of the population in 2017, which is a more than two-fold increase [4, 5]. The true burden of CKD in South Africa is unknown, owing to a lack of nationally representative studies in the country. Evidence suggests an increasing burden of non-communicable diseases in South Africa (hypertension, obesity and Type-2 diabetes mellitus) which are known risk factors for end-stage kidney disease (ESKD), particularly in black ethnic groups [6–8].

CKD often goes unnoticed because there are no specific symptoms, leading to delays in diagnosis or diagnosis at an advanced stage. Investigations for CKD are very simple and freely available in South Africa. The diagnosis of CKD is made in the presence of a persistently low glomerular filtration rate (GFR), which is estimated from the serum creatinine concentration or proteinuria over a period of at least three months. Evidence shows unequivocally that the Chronic Kidney Disease Epidemiology Collaboration Creatinine (CKD-EPI$_{Creatinine}$) equation is more accurate than the previous Modification of Diet for Renal Disease equation (MDRD) [2, 9]. The diagnostic challenges of CKD cannot be ignored, especially in rural communities where timeous access to relevant tests remains a concern. It has been estimated that 10% of the world's population has some degree of CKD, with trends suggesting an alarming increase in the incidence globally [10]. Current data estimates that about 5 million South Africans over the age of 20 years have CKD, and in black South Africans, the figure is almost certainly higher [3, 10]. In sub-Saharan Africa, the overall prevalence in the general population is 15,8% for CKD Stages 1–3 and 4.6% for Stages 3–5 in the general population [11].

By December 2012, there were 8 559 patients receiving chronic RRT in SA—6 952 on dialysis and 1 607 with a functioning kidney transplant. More than half of these patients were from the private sector, which serves less than 20% of the population. Facilities in the public sector, which not only serve more than 80% of the population but where the burden of Chronic Renal Failure (CRF) is about three times that in the private sector, are strictly limited. This means that only 15–20% of those who require RRT obtain such treatment because of limited resources. The approximate annual cost of dialysis is R200 000 per patient, and that of transplantation is R300 000 in the first year, and the cost rises to R160 000—R180 000 in subsequent years [10].

CKD complications represent a considerable burden on global healthcare resources, and only a few countries have sufficiently robust economies to meet the challenge posed by this disease [1]. As such, an intensified screening programme aimed at prioritising at-risk individuals at the population level might assist in early diagnosis and management to prevent further progression to end-stage kidney failure and mortality [4]. Without a nationally representative

study sample, it is difficult to gain a full understanding of the true burden of CKD in the country. Worse still, there is a paucity of published literature on CKD among people living in rural communities, especially in the Eastern Cape province. This has serious implications for crafting an effective national health promotion and disease prevention policy for the country. In order to contribute much-needed data on the burden of CKD among rural residents, which is currently unavailable in the country, this study determined the prevalence of kidney damage and examined the associated risk factors among individuals accessing care at the rural Mbekweni Community Health Centre in the Eastern Cape province, South Africa.

## Methods

### Ethical considerations

This study received ethical approval from the Research Ethics Committee of the Faculty of Medicine and Health Sciences, Walter Sisulu University (Reference Number: 025/2017). Permission for the study's implementation was granted by the Eastern Cape Department of Health, OR Tambo District Department of Health and the facility manager. Each participant gave written informed consent, indicating voluntary participation in the study. The study was implemented in accordance with the Helsinki Declaration and Good Clinical Practice Guidelines.

### Study design, setting and population

This observational cross-sectional study was conducted in the outpatient department of the Mbekweni Community Health Centre (CHC) in the King Sabata Dalindyebo (KSD) sub-district of OR Tambo district in the Eastern Cape province between May and July 2022. The Eastern Cape province of South Africa, specifically its north-eastern area, known as the wild coast, encompassing the district of Mbashe, is the most deprived area in the country, with a disproportionate burden of unemployment, poverty and disease [12, 13]. The area falls below national and regional standards for clean water, employment and access to healthcare services [12, 13]. The Eastern Cape covers 13.8% of the total area of the country and is home to 12.7% of the population, which utilises 10.2% of the domestic electrification and 6.5% of the domestic piped water [13, 14]. The incidence of infectious and chronic diseases, as well as malnutrition, is higher than the national average, and the coverage of immunisation and healthcare service delivery is the lowest [14].

The Mbekweni CHC serves the deeply rural residents of the Mbekweni location, providing health care services for patients with multi-morbidity, comprising an estimated population of 24 284 inhabitants predominantly of the black ethnic group [15]. The adult population (20 years and above) of Mbekweni is estimated to be 13 716 [15]. The sample size of 389 was estimated by using the free Software Epidat 3.1. Xunta de Galicia, Spain (Pan-American Health Organization/World Health Organization), using a confidence level of 95%, an expected proportion of 10%, and a maximum error of 3%.

In order to ensure inclusivity and minimise selection bias, participants were recruited through a simple random sampling technique; a hundred numbers were given to the adults attending the outpatient department of Mbekweni CHC, and 50 numbers were chosen randomly using the free Software Epidat 3.1. Xunta de Galicia Spain (Pan-American Health Organization/World Health Organization) run on Dr ERG's personal computer.

Participants were considered eligible if they were ≥ 20 years old at the time of the study, attending the outpatient department of Mbekweni CHC, and willing to participate in the research study. However, participants were excluded if they had acute medical (vomiting, diarrhoea, burns) or psychiatric emergencies that required urgent care at the health facility,

pregnancy, or any other limitations judged to interfere with study participation or their ability to follow study procedures (cognitive impairment, depression or psychotic disorders). In addition, first-degree relatives of participants were excluded.

To avoid bias, a 40 per cent was calculated over the estimated sample, a total of 545 participants were approached at the outpatient department during the study period. Of the total (N = 545), 112 declined participations (refused consent). A total of 44 participants were excluded for different reasons; 10 had a positive pregnancy test, who were promptly referred to the antenatal clinic, 10 with urinary tract infection were treated, eight patients with Dementia, and six with moderate to severe depression, five first degree relatives of the participants, and another five with acute diarrhoea.

## Study procedure

The lead author (ERG) and a research nurse trained in the study process implemented the protocol. The research nurse administered the interviewer-assisted questionnaire by reviewing the medical records and conducting the direct interviews, while ERG drew 5mL of venous blood sample from each consenting participant for additional investigation. Participants also submitted spot urine samples. Each questionnaire was issued a unique identifier code to link the participants' data to the laboratory investigation while maintaining privacy and confidentiality of medical information. The questionnaire comprised demographic information (age and sex), medical history (family history of hypertension and/or diabetes, current hypertension, current diabetes mellitus, current CKD and HIV) and laboratory results (urine dipstick for proteinuria, blood glucose, total cholesterol, low-density lipoprotein, high-density lipoprotein triglycerides, and serum creatinine). The blood pressure, weight and height of each participant was measured by the research nurse in accordance with standard protocols. All the participants with abnormal findings (proteinuria in the urine dipstick, decreased glomerular filtration as well as dyslipidaemia were followed up for further management at the health facility by ERG.

## Measures

Current kidney function was assessed by estimating the serum creatinine and confirming the presence of protein in urine samples. For the detection of proteinuria, spot urine samples were collected from the participants in specimen jars of 40mL. To avoid interpretation bias, the urine test strips were processed in a Urine Analysis Machine Model SLSSUA and interpreted as Protein Negative, Protein trace (15 mg/dL), Protein 1+ (30 mg/dL), Protein 2+ (100 mg/dL) or Protein 3+ (300 mg/dL).

To determine the serum creatinine, 5 mL of venous blood samples were drawn by ERG (a medical doctor) from each participant and sent on ice daily to the Nelson Mandela National Health Laboratory Services for processing, in accordance with standard protocols. The estimated glomerular filtration rate (eGFR) was estimated by using the Chronic Kidney Disease Epidemiology Collaboration Creatinine (CKD-EPI$_{creatinine}$) and the re-expressed four-variable Modified Diet in Renal Disease (MDRD) equations without any adjustment for black ethnicity. There is currently insufficient information in the literature about AmaXhosa people of South Africa and therefore, it is unclear which method is better at detecting kidney damage in all the diverse populations in the country [3, 16].

Participants were classified according to the Kidney Disease Outcomes Quality Initiatives CKD classification: GFR categories (ml/min/1.73m$^2$); G1 –Normal or High $\geq$ 90; G2 –Mildly Decreased (60–90); G3a –Mildly to Moderately Decreased (45–59); G3b –Moderately to Severely Decreased (30–44); G4 –Severely Decreased (15–29); G5 –Kidney Failure ($\leq$15) [17].

CKD was defined as an eGFR < 60ml/min/1.73 by CKD-Epi$_{creatinine}$ or the presence of kidney structural abnormalities for more than three months [18]. Hypertension (HTN) was defined as a persistent elevation of office blood pressure (BP) 140/90mmHg, in two or three readings several hours or days apart or a personal history of HTN and/or treatment with antihypertensive drugs [19]. Diabetes mellitus (DM) was defined as a persistent elevation of random blood sugar $\geq$ 11.1 mmol/l on two or more consecutive clinic visits or a history of diabetes and/or treatment with hypoglycaemic drugs [20].

## Statistical analysis

Data were analysed with the IBM SPSS Statistics for Windows, Version 27.0 (IBM Corp., Armonk, New York, USA). Descriptive statistics were used to summarise the baseline characteristics and were presented as counts and frequencies for categorical variables, means (± standard deviations) for normally distributed continuous variables, and medians (interquartile range) for non-normally distributed variables. The chi-squared test was used in the bivariate analysis to identify the association between baseline characteristics and CKD. Significant associations between the baseline characteristics and the main outcome (chronic kidney disease) were assessed by applying a multivariate logistic regression model analysis (both unadjusted and adjusted odds ratios) with a 95% confidence interval (95% CI). A p-value of less than 0.05 was considered statistically significant.

## Results

All 389 participants were black Africans, 69,9% of whom were females. The mean age of the participants was 52.3 (±17.5) years. The mean eGFR CK-EPI$_{creatinine}$ was 87,42 (±26.93) ml/min/1.73m$^2$. The prevalence of CKD (GFR categories G3–G5) was 17.22% (n = 67) as determined by CK-EPI$_{creatinine}$, and 17.73% (n = 69) as determined by the MDRD equation. Ten participants reported a prior diagnosis of CKD at the time of the study (Table 1).

**Table 1. Demographic and clinical characteristics of the participants.**

| Variables | |
|---|---|
| Age (Years) (n = 389) | 52.36 ± 17.5 |
| Sex (n = 389) | |
| Female | 272 (69.9) |
| Male | 117 (30.1) |
| Kidney function assessment (n = 389) | |
| Serum Creatinine(μmol/l) | 73 [26] |
| Serum Creatinine(mg/dl) | 0.80 [0.30] |
| Proteins (mg/dl) | 2 [7] |
| Estimated Glomerular Filtration rate (ml/min/1.73m$^2$) (n = 389) | |
| eGFR CKD-Epi$_{creatinine}$ | 87.42 ± 26.93 |
| eGFR CK MDRD | 89.08 ± 38.05 |
| Chronic Kidney Disease (CKD) (n = 389) | |
| eGFR CKD-Epi$_{creatinine}$ | 67 (17.22) |
| CKD (eGFR by MDRD) | 69 (17.73) |
| Participants with previous history of CKD | 10 |

n (%), Mean ± Standard deviation, Median [Interquartile range IQR] CKD-EPI$_{creatinine}$ = Chronic Kidney Disease Epidemiology Collaboration Creatinine, MDRD = Modified Diet in Renal Disease, CKD = chronic kidney disease, Source: Participants' questionnaire

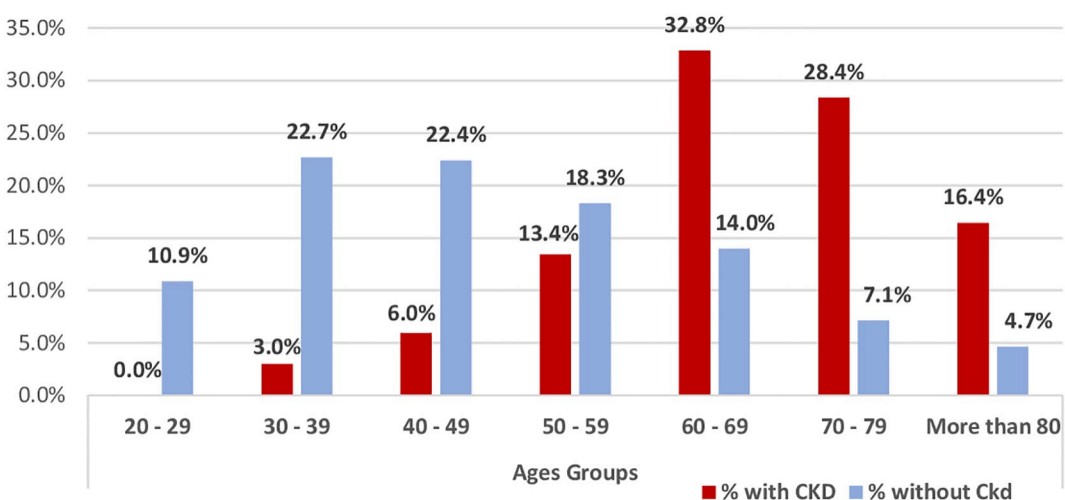

**Fig 1. Association between CKD-EPI$_{creatinine}$ and age categories.** Red box Percentage (%) of participants with CKD by the CKDEPI-creatinine values. Blue box Percentage (%) of patients without CKD by the CKDEPI-creatinine values.

### Associations of CKD with potential risk factors

There is a significant relationship between CKD and increasing age; prevalence ranged from 3% at 30–39 years to a maximum of 32.8% at 60–69 years, followed by 28.4% at 70–79 years to 16.4% at 80 years and over (p-value = 0.000) (Fig 1).

Table 2 shows the distribution of the participants with CKD grades from 3 to 5 by age group and sex.

Table 3 shows the distribution of the associated risk factors and patients with a previous CKD history by sex and CKD Grades 3 to 5.

In the bivariate analysis, the following potential risk factors were significantly associated with CKD: age (p<0.001), HTN (p<0.001), DM (p <0.001), HIV (p<0.001) and level of education (p<0.001). There were no sex differences in CKD prevalence (p = 0.529) (Table 4).

There was no association between the anti-hypertensive treatment and the CKD. At the moment of the study, four patients classified as grade 5 were on treatment with enalapril. It was stopped, and the patients were changed to amlodipine (Table 5).

**Table 2. Distribution of the CKD grades 3–5 (KDIGO classification) by age group and sex.**

| Age Groups | Grade 3 | | Grade 4 | | Grade 5 | | CKD (Total grades 3–5) | | Total |
|---|---|---|---|---|---|---|---|---|---|
| | F | M | F | M | F | M | F | M | |
| 20–29 | 0 | 0 | 0 | 0 | 0 | 0 | 0 (0.0) | 0 (0.0) | 0 (0.0) |
| 30–39 | 1 | 1 | 0 | 0 | 0 | 0 | 1 (2.0) | 1 (5.6) | 2 (3.0) |
| 40–49 | 1 | 1 | 0 | 0 | 2 | 0 | 3 (6.1) | 1 (5.6) | 4 (6.0) |
| 50–59 | 5 | 1 | 0 | 2 | 1 | 0 | 6 (12.2) | 3 (16.7) | 9 (13.4) |
| 60–69 | 15 | 3 | 4 | 0 | 0 | 0 | 19 (38.8) | 3 (16.7) | 22 (32.8) |
| 70–79 | 5 | 3 | 5 | 3 | 2 | 1 | 12 (24.5) | 7 (38.9) | 19 (28.4) |
| = > than 80 | 6 | 3 | 1 | 0 | 1 | 0 | 8 (16.3) | 3 (16.7) | 11 (16.4) |
| Total | 33 | 12 | 10 | 5 | 6 | 1 | 49 (73.1) | 18 (26.9) | 67 (17.2) |

n (%), Source: Participants' questionnaire

**Table 3. Distribution of the associated risk factors and previous CKD history by sex and CKD grades 3 to 5 (KDIGO classification).**

| Risk Factors | Grade 3 | | Grade 4 | | Grade 5 | | CKD stages Grades 3–5 | | CKD (n = 67) |
|---|---|---|---|---|---|---|---|---|---|
| | F | M | F | M | F | M | Total By Sex | | Total |
| | | | | | | | F | M | |
| HTN only | 18 (69.2) | 5 (71.4) | 6 (23.1) | 1 (14.3) | 2 (7.7) | 1 (14.3) | 26 (53.1) | 7 (38.1) | 33 (49.3) |
| DM only | 0 (0.0) | 1 (14.3) | 0 (0.0) | 0 (0.0) | 0 (0.0) | 0 (0.0) | 0 (0.0) | 1 (5.6) | 1 (1.5) |
| HIV only | 2 (7.7) | 2 (28.6) | 1 (3.8) | 0 (0.0) | 1 (3.8) | 0 (0.0) | 4 (8.2) | 2 (11.1) | 6 (9.0) |
| HTN + DM | 8 (30.8) | 1 (14.3) | 3 (11.5) | 2 (28.6) | 1 (3.8) | 0 (0.0) | 12 (24.5) | 3 (16.3) | 15 (22.4) |
| HTN + HIV | 1 (3.8) | 0 (0.0) | 0 (0.0) | 0 (0.0) | 0 (0.0) | 0 (0.0) | 1 (2.0) | 0 (0.0) | 1 (1.5) |
| HTN + DM + HIV | 1 (3.8) | 0 (0.0) | 0 (0.0) | 0 (0.0) | 0 (0.0) | 0 (0.0) | 1 (2.0) | 0 (0.0) | 1 (1.5) |
| Previous History of CKD | | | | | | | | | |
| HTN only | 0 (0.0) | 1 (14.3) | 0 (0.0) | 1 (14.3) | 0 (0.0) | 0 (0.0) | 0 (0.0) | 2 (11.1) | 2 (3.0) |
| HTN + DM | 3 (11.5) | 1 (14.3) | 0 (0.0) | 0 (0.0) | 1 (3.8) | 0 (0.0) | 4 (8.2) | 1 (5.6) | 5 (7.5) |
| HTN + HIV | 0 (0.0) | 1 (14.3) | 0 (0.0) | 0 (0.0) | 0 (0.0) | 0 (0.0) | 0 (0.0) | 1 (5.6) | 1 (1.5) |
| HTN + DM + HIV | 0 (0.0) | 0 (0.0) | 0 (0.0) | 1 (14.3) | 1 (3.8) | 0 (0.0) | 1 (2.0) | 1 (5.6) | 2 (3.0) |

HTN = Hypertension, HIV = Human Immunodeficiency Virus, DM = Diabetes Mellitus, Source: Participants' questionnaire

There was no association between diabetes treatment and CKD. At the time of the study, two patients classified as grade 5 were on treatment with Metformin, which was changed to Insulin (Table 6).

In the multiple logistic regression analysis, age as a continuous variable was significantly associated with the development of CKD in the study sample. Older age was identified as a significant risk factor for CKD, with an odds ratio (OR) = 1.08 (95% confidence interval [CI]: 1.06–1.1, $p < 0.001$) (Table 7).

Thirty-one participants (7.97%) had significant levels of protein in the urine ($\geq$ 30 mg/dl); their mean age was 56.68 (±15.45) years. In the bivariate analysis, HTN was significantly associated with proteinuria: HTN (p = 0.001) (OR = 4.17, 95% CI 1.67–10.4) (Table 8).

The study did not find any association between the treatment (as a risk factor) and the presence of proteinuria (Table 9).

## Discussion

Despite the increasing prevalence of hypertension, diabetes mellitus and other non-communicable diseases, all known risk factors for the development of CKD in South Africa, the true burden of CKD in the country is unknown. The study reports on the prevalence of CKD and examines some of its known risk factors among the residents of a rural community accessing Mbekweni Community Health Centre in the King Sabata Dalindyebo sub-district municipality of the Eastern Cape province. Findings from this study provide much-needed CKD data on the rural population of South Africa that may inform an effective kidney health prevention programme in the country.

**Table 4. Bivariate analysis showing risk factors between participants with and without CKD.**

| Variables | CKD | No CKD | Unadjusted Odds Ratios 95%(CI) | p-value |
|---|---|---|---|---|
| **Age** | 67.82±12.94 | 49.12±15.99 | 1.08 (1.06–1.1) | <0.001 |
| **Gender** | | | | 0.529 |
| Female (Ref) | 49(73.13) | 223(69.25) | 1 | |
| Male | 18 (26.87) | 99 (30.75) | 0.83 (0.46–1.49) | |
| **Hypertension** | 60 (89.55) | 144(44.72) | 10.6 (4.7–23.89) | <0.001 |
| **Diabetes mellitus** | 24 (35.82) | 56 (17.35) | 2.65 (1.49–4.72) | <0.001 |
| **HIV** | 11 (16.42) | 181 (56.21) | 6.54 (3.3–12.94) | <0.001 |
| **FHx of Hypertension** | 16(23.88) | 54(16.77) | 0.64 (0.34–1.21) | 0.17 |
| **FHx of Diabetes mellitus** | 10(14.93) | 39(12.11) | 0.79 (0.37–1.66) | 0.529 |
| **Alcohol consumption** | 5 (7.46) | 27 (8.71) | 0.88 (0.33–2.38) | 0.800 |
| **Current Smoking** | 2 (2.98) | 13 (4.03) | 0.73 (0.16–3.32) | 0.674 |
| **Level of Education** | | | | 0.000 |
| Illiterate (ref) | 14 (20.9) | 20 (6.21) | 1 | |
| Primary | 29 (43.28) | 101 (31.37) | 0.41 (0.19–0.91) | 0.029 |
| Secondary | 20 (29.85) | 197 (61.18) | 0.15 (0.06–0.33) | 0.000 |
| Tertiary | 4 (5.97) | 4 (1.24) | 1.43 (0.31–6.70) | 0.651 |
| **BMI** | 24.29 ± 5.27 | 23.76 ± 5.81 | 0.98 (0.94–1.03) | 0.489 |
| **Urine Protein(≥30mg/dl)** | 22 (32.8) | 9 (2.8) | 1.02 (1.01–1.03) | <0.001 |
| **Cholesterol** | 4.46 ± 0.91 | 4.42 ± 1.12 | 0.96 (0.73–1.28) | 0.802 |
| **Triglycerides** | 1.83 ± 1.01 | 1.59 ± 0.8 | 0.73 (0.44–1.23) | 0.236 |

HIV = Human immunodeficiency virus, BMI = Body mass index, CKD = Chronic kidney disease, FHx = Family History

The study found a prevalence of CKD of 17.2% in the sample, with no significant differences shown between the CKD-Epi$_{creatinine}$ and MDRD results. This prevalence (17.2%) is higher than the 6.1% reported by Adeniyi et al. among a cohort of teachers in an urban area of Cape Town in the Western Cape province in 2017 [9]. It should also be noted that in the Adeniyi et al. (2017) study, there was a considerable difference between the CKD results shown by the two equations; 6.1% according to MDRD and 1.8% according to CKD-EPI$_{creatinine}$ [9]. Similarly, the current study found a much higher prevalence of CKD than the 3.4% found by Peer et al in Cape Town in 2020 [21] and the 6.7% described by Fabian et al. in the rural area of Mpumalanga between 2017 and 2018 [22]. It should be noted that the CKD prevalence from Fabian et al. [22] was reported mainly for persistent albuminuria, which differs methodologically from the present study. The prevalence of 17.2% from the present study is lower than the 29.2% reported by Navise et al. in a community-based study from the North-West Province [23]. It is necessary to highlight that in Navise et al. [23], the crude prevalence is based on

**Table 5. Relationship between the Antihypertensive treatment and CKD grades (KDIGO classification).**

| Antihypertensive treatment | Grade 1 | Grade 2 | Grade 3 | Grade 4 | Grade 5 | Total Grades 1–2 | Total Grades 3–5 | Unadjusted Odds Ratio. CI (95%) | p-value |
|---|---|---|---|---|---|---|---|---|---|
| Amlodipine | 21 | 53 | 19 | 9 | 2 | 74 | 30 | 0.73(0.42–1.26) | 0.254 |
| Enalapril | 24 | 49 | 27 | 10 | 4 | 73 | 41 | 1.45 (0.83–2.53) | 0.189 |
| Diuretics | 45 | 86 | 38 | 12 | 6 | 131 | 56 | 1.68 (0.61–4.6) | 0.314 |
| Atenolol | 6 | 5 | 8 | 3 | 1 | 11 | 12 | 2.48(0.97–6.35) | 0.058 |

Source: Participants' questionnaire

**Table 6. Relationship between the Diabetes Mellitus treatment and CKD grades (KDIGO classification).**

| Diabetes Mellitus Treatment | Grade 1 | Grade 2 | Grade 3 | Grade 4 | Grade 5 | Total Grades 1–2 | Total Grades 3–5 | Unadjusted Odds Ratio. CI (95%) | p-value |
|---|---|---|---|---|---|---|---|---|---|
| Metformin | 18 | 32 | 14 | 4 | 2 | 50 | 20 | 3.37 (0.67–17.01) | 0.196 |
| Sulfonylureas | 9 | 11 | 9 | 3 | 0 | 20 | 12 | 0.57 (0.23–1.43) | 0.232 |
| Insulin | 6 | 7 | 2 | 2 | 1 | 13 | 5 | 1.21 (0.41–3.54) | 0.793 |

Source: Participants' questionnaire

**Table 7. Multiple logistic regression analysis of risk factors associated with CKD by CKD-Epi$_{creatinine}$.**

| Variables | Adjusted Odds Ratios (95% CI) | p-value |
|---|---|---|
| Age | 1.08 (1.06–1.1) | <0.001 |
| Hypertension | 0.48 (0.16–1.4) | 0.179 |
| Diabetes mellitus | 0.64 (0.33–1.23) | 0.182 |
| HIV | 0.78 (0.3–1.98) | 0.595 |
| Level of Education | | |
| Illiterate (ref) | 1 | |
| Primary | 1.3 (0.54–3.13) | 0.555 |
| Secondary | 1.28 (0.48–3.39) | 0.617 |

HIV = Human immunodeficiency virus.

**Table 8. Bivariate analysis showing association between potential risk factors and proteinuria.**

| Variables | Urine Proteins < 30 mg/dl (n = 358) | Urine Proteins ≥30 mg/dl (n = 31) | Unadjusted Odds Ratios. CI (95%) | p-value |
|---|---|---|---|---|
| Age | 51.97±17.13 | 56.68 ± 15.45 | 1.02 (0.99–1.04) | 0.141 |
| Sex | | | | 0.894 |
| Female (ref) | 250 (69.83) | 22 (70.97) | 1.06 (0.47–2.37) | |
| Male | 108 (30.17) | 9 (29.03) | | |
| Hypertension | 179 (50.0) | 25 (80.65) | 4.17 (1.67–10.4) | 0.001 |
| Diabetes mellitus | 22 (70.97) | 9 (29.03) | 1.65 (0.73–3.75) | 0.243 |
| HIV | 182 (50.84) | 10 (32.26)) | 0.46 (0.21–1.01) | 0.52 |
| BMI | 23.8 ± 5.66 | 24.21 ± 6.38 | 1.01 (0.95–1.08) | 0.707 |
| Cholesterol | 4.4 ± 1.07 | 4.74 ± 1 | 1.32 (0.91–1.92) | 0.148 |
| Triglycerides | 1.67 ± 0.9 | 1.56 ± 0.55 | 1.18 (0.55–2.53) | 0.674 |

HIV = Human immunodeficiency virus; BMI = Body mass index. Source: Participants' questionnaire. Data expressed as number (percentage) and mean ± standard deviation. Urine strip results from urine analysis machine model SLSSUA.

an eGFR < 90 ml/min/1.73m$^2$. The prevalence for the eGFR < 60 ml/min/1.73m$^2$ is 3% lower than in our study. The CKD definition differs from the one used in our study. However, the prevalence of CKD in the current study is similar to the 17.3% reported by Matsha et al in Cape Town in 2013 [24]. The wide disparity in the prevalence of CKD can be explained by the peculiarities of each study population. While the participants in the current study were older (mean age 52.36 years) and had a high prevalence of hypertension (52.4%), diabetes mellitus (20.6%) and other cardiovascular risk factors, similar to the participants in Matsha et al in Cape Town [24], relatively younger (mean age 44.1–46.3 years) and more active cohorts, with

**Table 9. Relationship between the treatment and proteinuria.**

| Antihypertensive treatment | Urine Proteins | | p-value | Unadjusted Odds Ratio. CI (95%) |
|---|---|---|---|---|
| | Equal or more than 30 mg/dl | Less than 30 mg/dl | | |
| Amlodipine | 7 | 97 | 0.866 | 0.92(0.34–2.48) |
| Enalapril | 12 | 102 | 0.398 | 0.7 (0.3–1.61) |
| Atenolol | 4 | 19 | 0.415 | 0.56(0.15–2.11) |
| Diuretics | 22 | 165 | 0.482 | 0.62 (0.17–2.34) |
| Diabetes treatment | | | | |
| Metformin | 7 | 63 | 0.301 | 0.3(0.03–2.95) |
| Sulfonylureas | 4 | 28 | 0.913 | 1.09 (0.23–5.16) |
| Insulin | 2 | 16 | 0.622 | 0.55(0.05–0.598) |

a lower prevalence of cardiovascular disease, were included in Booysen et al [16], Adeniyi et al [9] Peer et al [21] and Navise et al. [23], the median age in the Fabian et al. study was 35 years [22].

Several studies have compared different glomerular filtration rate equations measured by nuclear medicine methods [16, 21, 25]. The authors suggest that more studies are needed to validate these equations in the diverse local populations of South Africa. Booysen et al. [16] acknowledged that the CKD-Epi$_{creatinine}$ equation may not be as accurate for eGFR assessments among black Africans as it is among the white population. However, this equation represents the most appropriate equation for detecting pre-clinical cardiac and vascular end-organ damage beyond the conventional risk factors found in black Africans [13]. Moreover, Holnes et al. [26] reported that both the MDRD and CKD-EPI equations have shown satisfactory accuracy in the South African mixed-ancestry adult population.

In the current study, female participants were predominant (69.9%), which is not surprising, given that this cohort was drawn from the local community health centre. This is in keeping with other studies in South Africa, where the percentages of female participants were 70.3% [3], 64.1% [21], 58% [22], 62,8% [23] and 75.2% [24]. There are few studies in rural South Africa to compare. However, a study by Kaze et al. in Cameroon found a slightly higher male predominance at 53.4% [27]. Although Matsha et al. [24] reported a significant relationship between female sex and CKD, the current study found no significant association with sex. This is in agreement with Peer et al. [21], Fabian et al. [22], and Kaze et al. [27], whose studies demonstrated no significant association between sex and CKD. The impact of sex on CKD remains controversial; although some studies document a higher risk of developing CKD among women [23, 24, 28, 29], others report higher odds among men [30–33].

Previous studies have reported many risk factors for the development of CKD in different population groups in South Africa, especially in the urban parts of Cape Town [9, 21, 24]. However, such risk factors have not been sufficiently investigated in the rural communities of South Africa, especially in the Eastern Cape province. The current study showed a significant association between CKD and the ageing population. The mean age of the participants with CKD was 67.82 years, with those above 60 years most affected. This finding corroborates the findings of Peer et al. [21] and Kaze et al [27]. However, Adeniyi et al. [9] reported the presence of CKD in younger age groups (mean age 47.3 years) among teachers in Cape Town.

The current study found a proteinuria prevalence of 7.97%, which was significantly associated with the presence of hypertension. A similar prevalence of proteinuria (7.2%) and association with hypertension was reported by Kaze et al. [27]. However, a lower prevalence of proteinuria (4.5%) in a younger cohort of teachers was reported by Adeniyi et al [9]. It should

be pointed out that the urine—protein creatinine ratio was not assessed in the current study, as it was in Adeniyi et al. study [9]. Although other studies report higher rates of proteinuria among women than men [3, 28, 33], the current study found no association with sex. In other studies, proteinuria was not evaluated or was related to groups of participants with specific co-morbidities such as HIV ART naïve, HIV on ART, hypertension, or only diabetes mellitus [30, 34–39].

## Study limitations

First, this is a facility-based study with respondents recruited from the outpatient clinic thus, limits the ability to generalise the findings to the general population. In addition, the study population consists exclusively of black South Africans from the rural KSD sub-district, limiting the applicability of the findings to the diverse ethnicities of the rural South African population.

In addition, the cohort's predominance of females reflects a common trend in health service utilisation in South Africa, as elucidated in many population studies in the country [3, 21, 24]. Notwithstanding, there was no difference by sex in the CKD prevalence in this study. Future studies on CKD prevalence should target the broader rural population of the country. Furthermore, creatinine was measured using an assay based on the Jaffe reaction, which is more susceptible to interferences than the enzymatic method.

Finally, the use of the CKD-EPI$_{creatinine}$ equation for the detection of CKD needs further validation in the local context, even though both the MDRD and CKD-EPI equations have shown a satisfactory performance in the South African mixed-ancestry adult population. The cross-sectional nature of the study design did not allow for chronicity and might have overestimated chronic kidney disease prevalence in the setting. A previous study showed an overestimation of CKD in an older population and an underestimation in a younger population [40]. Although individuals with probable acute kidney injuries (vomiting, diarrhoea and burns) were excluded from the study, a repeat measurement of creatinine and proteinuria at three months would have established the diagnosis of chronic kidney disease.

## Conclusions

This study found a high prevalence of CKD (17.2%) and proteinuria (7.97%) in this rural community, largely attributed to advanced age and hypertension, respectively. This finding supports the ideal hospital framework of the National Department of Health in South Africa, which focuses on screening for cardiovascular and other non-communicable diseases at every facility. Early detection of proteinuria and decreased renal function at community health centres should trigger a referral to a higher level of care for further management of patients.

## Acknowledgments

The authors thank the doctors and nurses of the Mbekweni Community Health Centre for their support during the study.

## Author Contributions

**Conceptualization:** Ernesto Rosales Gonzalez, Parimalanie Yogeswaran, Guillermo Alfredo Pulido Estrada.

**Data curation:** Ernesto Rosales Gonzalez, Guillermo Alfredo Pulido Estrada.

**Formal analysis:** Ernesto Rosales Gonzalez, Parimalanie Yogeswaran, Guillermo Alfredo Pulido Estrada.

**Investigation:** Ernesto Rosales Gonzalez.

**Methodology:** Ernesto Rosales Gonzalez, Parimalanie Yogeswaran, Jimmy Chandia, Guillermo Alfredo Pulido Estrada.

**Supervision:** Jimmy Chandia, Oladele Vincent Adeniyi.

**Writing – original draft:** Ernesto Rosales Gonzalez.

**Writing – review & editing:** Jimmy Chandia, Oladele Vincent Adeniyi.

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
