## [Decision Letter · Decision Letter 0]

30 Oct 2023

PONE-D-23-27552Kidney Damage and Associated Risk Factors in the Rural Eastern Cape, South Africa: A Cross-Sectional StudyPLOS ONE

Dear Dr. Rosales Gonzalez,

Thank you for submitting your manuscript to PLOS ONE. After careful consideration, we feel that it has merit but does not fully meet PLOS ONE’s publication criteria as it currently stands. Therefore, we invite you to submit a revised version of the manuscript that addresses the points raised during the review process.

We look forward to receiving your revised manuscript.

Kind regards,

Donovan Anthony McGrowder, PhD., MA., MSc

Academic Editor

PLOS ONE

Journal Requirements:

Additional Editor Comments:

Your manuscript “Kidney Damage and Associated Risk Factors in the Rural Eastern Cape, South Africa: A Cross-Sectional Study” has been assessed by our reviewers. They have raised a number of points which we believe would improve the manuscript and may allow a revised version to be published in PLOS ONE. Their reports, together with any other comments, are below.

 If you are able to fully address these points, we would encourage you to submit a revised manuscript to PLOS ONE.

Reviewers' comments:

Reviewer's Responses to Questions

**Comments to the Author**

1. Is the manuscript technically sound, and do the data support the conclusions?

Reviewer #1: Partly

Reviewer #2: No

Reviewer #3: Yes

Reviewer #4: Yes

Reviewer #5: Partly

2. Has the statistical analysis been performed appropriately and rigorously? 

Reviewer #1: Yes

Reviewer #2: No

Reviewer #3: Yes

Reviewer #4: No

Reviewer #5: Yes

3. Have the authors made all data underlying the findings in their manuscript fully available?

Reviewer #1: No

Reviewer #2: Yes

Reviewer #3: No

Reviewer #4: Yes

Reviewer #5: Yes

4. Is the manuscript presented in an intelligible fashion and written in standard English?

Reviewer #1: No

Reviewer #2: Yes

Reviewer #3: Yes

Reviewer #4: Yes

Reviewer #5: Yes

5. Review Comments to the Author

Reviewer #1: The concept of this study is interesting. However, this study examined only a limited number cases in a certain region. I think it would be better to submit this paper to a journal of relevant regions.

1. It is necessary to show in more detail the clinical and social background of the regions.

2. It is necessary to make comparisons with other regions and clarify the characteristics of the target region.

3. Information about treatment for HT, DM and CKD is needed.

Reviewer #2: The authors investigated the prevalence of CKD in the Rural Eastern Cape, South Africa. Although the findings were partially meaningful, there are several serious problems. Especially, it is not surprising that both older age and the presence of proteinuria are strongly associated with the presence of CKD. Although they aimed to clarify the prevalence of CKD in this area, they included only outpatients of the hospitals. In contrast, the study by Adeniyi AB et al. [Ref. 9] included young school teachers, and the prevalence of CKD was lower as compared to this study. From this study design, they cannot conclude that early detection of proteinuria and decreased renal function could lead to prompt initiation of preventative measures and management to delay the progression to end-stage kidney failure and mortality.

Abstract

P.2, L.15: The authors described that significant kidney damage was defined as low eGFR (<60mL/min per 1.73m2) and/or the presence of proteinuria. However, the chief outcome of this study seems the prevalence of CKD defined by KDIGO Guideline.

P.2, L.21: Although the authors described that older age and the presence of proteinuria were associated with the significant kidney damage, both ORs were less than 1.

Introduction

P.3, L.30-32: Abbreviations of RRT should be spelled out at first usage and the abbreviated form used thereafter. Please check all abbreviations including CKD, eGFR, CKD-EPIcreatinine or MDRD.

Methods

P.8, L.133: Ref.14 is guideline for the management of glomerular diseases. CKD should be defined by KDIGO guideline [Levey AS, et al. Kidney Int 2005；67：2089‒100].

Results

P.9, L.156 and Table 1: The authors described that ten participants reported prior diagnosis of CKD at the time of the study (Table 1). However, there was no description in Table 1. There is no need to present the number and proportion of both female and male participants. Not-normally distributed variables, especially serum creatinine and urine protein, should be presented as medians and ranges or interquartile ranges (IQR).

P.11, Table 2: The number of HIV patients without CKD is odd. Was the prevalence of CKD higher among HIV patients? Urine protein of CKD and non-CKD were odd.

P.12, Table 3: ORs of high age and urine protein should be more than 1. P-value of diabetic mellitus is odd. Although the authors described the abbreviations of HTN, DM, eGFR and CKD-EPI below the Table 3, they did not use in the Table.

P.13, Table 4: The data of both patients with urine protein >30 and those with urine protein<30 should be presented. Although the OR of HIV is 0.46 (0.21-1.01), p-value was less than 0,05. OR of triglyceride is odd.

P.14, Table 5: The objective variable of this multivariate analysis is unclear. It is not much meaning to find that hypertension and higher serum creatinine level were associated with higher prevalence of proteinuria.

Reviewer #3: This cross-sectional study describes the prevalence of CKD in patients attending an outpatient clinic of a rural community in the eastern cape of South Africa. The article is well written and easy to understand. There are some limitations to the study including the homogeneity of the study population, the fact that this study was conducted in patients attending the out-patient department of a community clinic but these have been mentioned by the authors.

There are a few clarifications which should be addressed.

1. Please describe the outpatient department in more detail. Does it provide wellness and preventive services or it caters to patients who are being followed up for several illnesses? Although the results suggest the latter, this should be clearly stated in the methodology.

2. References are not always well written, some references do not have year of publications, and different styles have been used. Please refer to Plos one referencing style.

3. Some of the references do not seem relevant to the text, for example ref 19 refers to prevalence of CKD in Cameroonians. References should be sought from South African patients while discussing the demographics.

Reference 27 refers to precision nephrology, it may be better to quote the article quoted in that write-up: Minutolo R et al... Sex differences in the progression of CKD.......

In Conclusion, the study adds to the body of knowledge on prevalence of CKD to a subset of South Africans. Minor corrections noted above should be made.

Reviewer #4: This observational study investigates the prevalence of kidney damage within the rural community of the Eastern Cape province in South Africa. Notably, the research delves into the association of kidney damage with various risk factors linked to Chronic Kidney Disease (CKD). This study serves as a crucial endeavor in understanding the prevalence of CKD within the context of rural South Africa.

However, upon careful review, certain aspects require further elucidation and refinement.

1. Firstly, it is imperative to delineate the relationship between kidney function and specific factors, such as diabetes, hypertension, and proteinuria, differentiating the data based on gender. Analyzing these associations separately for males and females could provide a more comprehensive understanding of the impact of these factors on kidney health.

2. Moreover, the report highlights that 67 patients were classified between G3-G5 stages of CKD. However, the distribution of these patients in terms of gender remains ambiguous. It is essential to clarify the proportion or percentage of male and female patients within these stages for a more nuanced interpretation of the data.

3. Additionally, exploring the relationship between the grades of CKD and associated risk factors might offer deeper insights into the progression of the disease and its underlying causes. By examining how different risk factors correspond with the severity of CKD, a clearer understanding of the disease's trajectory can be obtained.

4. In reference to Table 2, while the data for Female CKD (49) and No-CKD (223) is presented, the information pertaining to males seems to be absent. Clarification is required regarding whether Table 2 is exclusively related to female subjects or if the data for male subjects was inadvertently omitted.

5. Lastly, in the paragraph discussing the 'Associations of CKD with potential risk factors,' an apparent inconsistency is noted regarding the relationship between age and the incidence of CKD. Contrary to the initial suggestion that the incidence of CKD declines with age, Figure 1 clearly illustrates an increasing ratio of CKD to Non-CKD cases with advancing age. This inconsistency should be rectified to ensure accurate interpretation and reporting of the data.

Reviewer #5: This is an interesting study as very few data exist about CKD in rural areas of South Africa.

The authors should be congratulated for their efforts to investigate CKD epidemiology in a region with shortages of health facilities.

However, I do have some comments.

1. The authors should give more information about the region. How much remoted is this? What are the main demographic data of the whole population and study group (employment, length of life, poverty, educational status, access to health facilities and so on)

2. There are no data about smoking and alcohol.

3. There is a possibility of selection bias as the participants were outpatients of the Health Centre. The authors should comment on this.

4. The authors should give a chart about the initial number of persons asked to participate. How many people refused or were excluded?

5. I miss information about the questionnaire given to patients and how it was filled in (assisted? , could all patients read?)

6. In the discussion, the authors could add comments/comparisons with other South African rural areas data ( Navise et al. https://doi.org/10.1186/s12882-023-03068-7 June Fabian et al https://doi.org/10.12688/wellcomeopenres.18016.2)

6. PLOS authors have the option to publish the peer review history of their article (what does this mean?). If published, this will include your full peer review and any attached files.

Reviewer #1: No

Reviewer #2: No

Reviewer #3: **Yes: **Ngozi Virginia Aikpokpo

Reviewer #4: No

Reviewer #5: No

---

## [Author Response · Author response to Decision Letter 0]

8 Feb 2024

Dear Editor,

 Thanks for the insightful comments of the reviewers. Indeed, the manuscript has improved remarkably with the suggestions and queries from the reviewers. Please find below the responses to the reviewers’ comments. 

Regards

Ernesto Rosales Gonzalez 

Reviewer #1: 

1. It is necessary to show in more detail the clinical and social background of the regions.

Response: Thanks for the comment. The clinical and social background has been included from lines 54 to 61 and 88 to 95.

2. It is necessary to make comparisons with other regions and clarify the characteristics of the target region.

Response: Thanks for the comment. We have added more information on the study setting. 

3. Information about treatment for HT, DM and CKD is needed.

Response: Thanks for the comment. We have made corrections

Reviewer #2: 

1. The authors investigated the prevalence of CKD in the Rural Eastern Cape, South Africa. Although the findings were partially meaningful, there are several serious problems. Especially, it is not surprising that both older age and the presence of proteinuria are strongly associated with the presence of CKD. Although they aimed to clarify the prevalence of CKD in this area, they included only outpatients of the hospitals. 

Response: Thanks for the comment. It should be clarified that the study comprised of stable patients attending the primary health care outpatient setting. This is the first point of contact with patients in the rural South African settings. More studies are needed to elucidate on the in-patient in rural hospital settings in South Africa.

2. In contrast, the study by Adeniyi AB et al. [Ref. 9] included young school teachers, and the prevalence of CKD was lower as compared to this study. From this study design, they cannot conclude that early detection of proteinuria and decreased renal function could lead to prompt initiation of preventative measures and management to delay the progression to end-stage kidney failure and mortality.

Response: Thanks for the insight. We have rephrase our conclusion on this.

Abstract

P.2, L.15: The authors described that significant kidney damage was defined as low eGFR (<60mL/min per 1.73m2) and/or the presence of proteinuria. However, the chief outcome of this study seems to be the prevalence of CKD defined by KDIGO Guideline.

Response: Thanks for the comment. We have made corrections 

P.2, L.21: Although the authors described that older age and the presence of proteinuria were associated with the significant kidney damage, both ORs were less than 1.

Response: Thanks for the comment. We have made corrections.

Introduction

P.3, L.30-32: Abbreviations of RRT should be spelled out at first usage and the abbreviated form used thereafter. Please check all abbreviations including CKD, eGFR, CKD-EPIcreatinine or MDRD.

Response: Thanks for the comment. We have made corrections as suggested

Methods

P.8, L.133: Ref.14 is guideline for the management of glomerular diseases. CKD should be defined by KDIGO guideline [Levey AS, et al. Kidney Int 2005；67：2089‒100].

Response: Thanks for the comment. We have made corrections as suggested

Results

P.9, L.156 and Table 1: The authors described that ten participants reported prior diagnosis of CKD at the time of the study (Table 1). However, there was no description in Table 1. There is no need to present the number and proportion of both female and male participants. Not-normally distributed variables, especially serum creatinine and urine protein, should be presented as medians and ranges or interquartile ranges (IQR).

Response: Thanks for the comment. We have made corrections as suggested

P.11, Table 2: The number of HIV patients without CKD is odd. Was the prevalence of CKD higher among HIV patients? Urine protein of CKD and non-CKD were odd.

Response: Thanks for the comment. We have made corrections as suggested

P.12, Table 3: ORs of high age and urine protein should be more than 1. P-value of diabetic mellitus is odd. Although the authors described the abbreviations of HTN, DM, eGFR and CKD-EPI below the Table 3, they did not use in the Table.

Response: Thanks for the comment. We have made corrections as suggested

P.13, Table 4: The data of both patients with urine protein >30 and those with urine protein<30 should be presented. Although the OR of HIV is 0.46 (0.21-1.01), p-value was less than 0,05. OR of triglyceride is odd.

Response: Thanks for the comment. We have made corrections as suggested

P.14, Table 5: The objective variable of this multivariate analysis is unclear. It is not much meaning to find that hypertension and higher serum creatinine level were associated with higher prevalence of proteinuria.

Response: Thanks for the comment. We have made corrections as suggested

Reviewer #3: 

This cross-sectional study describes the prevalence of CKD in patients attending an outpatient clinic of a rural community in the eastern cape of South Africa. The article is well written and easy to understand. There are some limitations to the study including the homogeneity of the study population, the fact that this study was conducted in patients attending the out-patient department of a community clinic but these have been mentioned by the authors.

There are a few clarifications which should be addressed.

1. Please describe the outpatient department in more detail. Does it provide wellness and preventive services or it caters to patients who are being followed up for several illnesses? Although the results suggest the latter, this should be clearly stated in the methodology.

Response: Thanks for the comment. We have made corrections

2. References are not always well written, some references do not have year of publications, and different styles have been used. Please refer to Plos one referencing style.

Response: Thanks for the comment. We have made corrections as suggested

3. Some of the references do not seem relevant to the text, for example ref 19 refers to prevalence of CKD in Cameroonians. References should be sought from South African patients while discussing the demographics.

Reference 27 refers to precision nephrology, it may be better to quote the article quoted in that write-up: Minutolo R et al... Sex differences in the progression of CKD.......

In Conclusion, the study adds to the body of knowledge on prevalence of CKD to a subset of South Africans. Minor corrections noted above should be made.

Response: Thanks for the comment. We have made corrections as suggested.

Reviewer #4: 

This observational study investigates the prevalence of kidney damage within the rural community of the Eastern Cape province in South Africa. Notably, the research delves into the association of kidney damage with various risk factors linked to Chronic Kidney Disease (CKD). This study serves as a crucial endeavor in understanding the prevalence of CKD within the context of rural South Africa.

However, upon careful review, certain aspects require further elucidation and refinement.

1. Firstly, it is imperative to delineate the relationship between kidney function and specific factors, such as diabetes, hypertension, and proteinuria, differentiating the data based on gender. Analyzing these associations separately for males and females could provide a more comprehensive understanding of the impact of these factors on kidney health.

Response: Thanks for the comment. We have made corrections as suggested

2. Moreover, the report highlights that 67 patients were classified between G3-G5 stages of CKD. However, the distribution of these patients in terms of gender remains ambiguous. It is essential to clarify the proportion or percentage of male and female patients within these stages for a more nuanced interpretation of the data.

Response: Thanks for the comment. We have made corrections as suggested

3. Additionally, exploring the relationship between the grades of CKD and associated risk factors might offer deeper insights into the progression of the disease and its underlying causes. By examining how different risk factors correspond with the severity of CKD, a clearer understanding of the disease's trajectory can be obtained.

Response: Thanks for the comment. We have made corrections as suggested

4. In reference to Table 2, while the data for Female CKD (49) and No-CKD (223) is presented, the information pertaining to males seems to be absent. Clarification is required regarding whether Table 2 is exclusively related to female subjects or if the data for male subjects was inadvertently omitted.

Response: Thanks for the comment. We have made corrections as suggested

5. Lastly, in the paragraph discussing the 'Associations of CKD with potential risk factors,' an apparent inconsistency is noted regarding the relationship between age and the incidence of CKD. Contrary to the initial suggestion that the incidence of CKD declines with age, Figure 1 clearly illustrates an increasing ratio of CKD to Non-CKD cases with advancing age. This inconsistency should be rectified to ensure accurate interpretation and reporting of the data.

Response: Thanks for the comment. We have made corrections as suggested

Reviewer #5: 

This is an interesting study as very few data exist about CKD in rural areas of South Africa. The authors should be congratulated for their efforts to investigate CKD epidemiology in a region with shortages of health facilities.

However, I do have some comments.

1. The authors should give more information about the region. How much remoted is this? What are the main demographic data of the whole population and study group (employment, length of life, poverty, educational status, access to health facilities and so on.

Response: Thanks for the comment. We have made corrections as suggested.

2. There are no data about smoking and alcohol.

Response: Thanks for the comment. We have made corrections as suggested

3. There is a possibility of selection bias as the participants were outpatients of the Health Centre. The authors should comment on this.

Response: Thanks for the comment. This primary health care study is the first in the Eastern Cape province, however, future studies should focus on CKD at the community level. 

4. The authors should give a chart about the initial number of persons asked to participate. How many people refused or were excluded?

Response: Thanks for the comment. We have made corrections as suggested

5. I miss information about the questionnaire given to patients and how it was filled in (assisted? , could all patients read?)

Response: Thanks for the comment. The research nurse provided assistance to the participants when required (line 107)

6. In the discussion, the authors could add comments/comparisons with other South African rural areas data ( Navise et al. https://doi.org/10.1186/s12882-023-03068-7 June Fabian et al https://doi.org/10.12688/wellcomeopenres.18016.2). 

Response: Thanks for the comment. We have made corrections as suggested

This observational study investigates the prevalence of kidney damage within the rural community of the Eastern Cape province in South Africa. Notably, the research delves into the association of kidney damage with various risk factors linked to Chronic Kidney Disease (CKD). This study serves as a crucial endeavor in understanding the prevalence of CKD within the context of rural South Africa. However, upon careful review, certain aspects require further elucidation and refinement.

1. Firstly, it is imperative to delineate the relationship between kidney function and specific factors, such as diabetes, hypertension, and proteinuria, differentiating the data based on gender. Analyzing these associations separately for males and females could provide a more comprehensive understanding of the impact of these factors on kidney health.

Response: Thanks for the comment. We have made corrections as suggested

2. Moreover, the report highlights that 67 patients were classified between G3-G5 stages of CKD. However, the distribution of these patients in terms of gender remains ambiguous. It is essential to clarify the proportion or percentage of male and female patients within these stages for a more nuanced interpretation of the data.

Response: Thanks for the comment. We have made corrections as suggested

3. Additionally, exploring the relationship between the grades of CKD and associated risk factors might offer deeper insights into the progression of the disease and its underlying causes. By examining how different risk factors correspond with the severity of CKD, a clearer understanding of the disease's trajectory can be obtained.

Response: Thanks for the comment. We have made corrections as suggested

4. In reference to Table 2, while the data for Female CKD (49) and No-CKD (223) is presented, the information pertaining to males seems to be absent. Clarification is required regarding whether Table 2 is exclusively related to female subjects or if the data for male subjects was inadvertently omitted.

Response: Thanks for the comment. We have made corrections as suggested

5. Lastly, in the paragraph discussing the 'Associations of CKD with potential risk factors,' an apparent inconsistency is noted regarding the relationship between age and the incidence of CKD. Contrary to the initial suggestion that the incidence of CKD declines with age, Figure 1 clearly illustrates an increasing ratio of CKD to Non-CKD cases with advancing age. This inconsistency should be rectified to ensure accurate interpretation and reporting of the data.

Response: Thanks for the comment. We have made corrections as suggested

---

## [Decision Letter · Decision Letter 1]

27 May 2024

PONE-D-23-27552R1Kidney Damage and Associated Risk Factors in the Rural Eastern Cape, South Africa: A Cross-Sectional StudyPLOS ONE

Dear Dr. Rosales Gonzalez,

Thank you for submitting your manuscript to PLOS ONE. After careful consideration, we feel that it has merit but does not fully meet PLOS ONE’s publication criteria as it currently stands. Therefore, we invite you to submit a revised version of the manuscript that addresses the points raised during the review process.

We look forward to receiving your revised manuscript.

Kind regards,

Ibrahim Sebutu Bello, MBBS, MPH, MD, FMCGP

Academic Editor

PLOS ONE

Journal Requirements:

Additional Editor Comments:

I want to commend the authors for their work on this important topic and for the diligence with which they have incorporated the reviewers' corrections and recommendations. Please kindly see below a few pending issues that need to be sorted out before we can proceed to the final acceptance of the manuscript.

General Comment:

The candidate uploaded the corrected manuscript to the manuscript with Track changes but not to the unmarked manuscript with no Track changes. Table 6 needs to be included in the corrected manuscript, and there was no corresponding correction in the abstract section. Hence, the corrected Manuscript in the unmarked should be uploaded, the table numbering corrected, and the abstract should reflect the changes made in the new tables 8.

Abstract: Page 2, Line 22-23 - Based on the changes the authors have effected in the body of the manuscript, the following statement should be corrected: “Risk factors for significant kidney damage were older age (OR=1.08, 95% CI 1.06-1.1, p<0.001)

Specifically, I request that you review the abstract to ensure it accurately reflects the changes made in the manuscript. This includes the correction of the statement regarding the risk factors for significant kidney damage.

Page 15, line 289: In line with reviewers' comments, the authors have changed the data points for Age on Table 8 (Multiple regression). Please make the same changes in the text and in the abstract. Similarly, the authors have removed urine protein from the table. Please make the same change in the abstract.

Table 8: Please clarify if the variable ‘Age’ as used in the logistic regression models was a continuous or categorical variable.

Limitations: Page 20, line 392—Being a facility-based study with respondents recruited from the outpatient clinic limits the ability to generalise the findings to the general population. Please kindly indicate this fact as part of the study limitations.

References: Please kindly ensure that all the references are in alignment with the journal requirements

Once these issues are addressed, we anticipate being able to accept the manuscript for publication. We kindly request that you resubmit as soon as possible. Your prompt action will greatly assist in expediting the publication process. Thank you.

Reviewers' comments:

Reviewer's Responses to Questions

**Comments to the Author**

1. If the authors have adequately addressed your comments raised in a previous round of review and you feel that this manuscript is now acceptable for publication, you may indicate that here to bypass the “Comments to the Author” section, enter your conflict of interest statement in the “Confidential to Editor” section, and submit your "Accept" recommendation.

Reviewer #1: All comments have been addressed

Reviewer #3: All comments have been addressed

Reviewer #4: All comments have been addressed

Reviewer #5: All comments have been addressed

2. Is the manuscript technically sound, and do the data support the conclusions?

Reviewer #1: Yes

Reviewer #3: Yes

Reviewer #4: Yes

Reviewer #5: No

3. Has the statistical analysis been performed appropriately and rigorously? 

Reviewer #1: I Don't Know

Reviewer #3: I Don't Know

Reviewer #4: Yes

Reviewer #5: Yes

4. Have the authors made all data underlying the findings in their manuscript fully available?

Reviewer #1: Yes

Reviewer #3: No

Reviewer #4: Yes

Reviewer #5: Yes

5. Is the manuscript presented in an intelligible fashion and written in standard English?

Reviewer #1: Yes

Reviewer #3: Yes

Reviewer #4: (No Response)

Reviewer #5: Yes

6. Review Comments to the Author

Reviewer #1: (No Response)

Reviewer #3: The authors have addressed all my comments. The limitations of the study are clearly stated and the need for further studies involving a wider range of participants have been noted.

Reviewer #4: (No Response)

Reviewer #5: I have no more comments.

............................................................... ..............................................

7. PLOS authors have the option to publish the peer review history of their article (what does this mean?). If published, this will include your full peer review and any attached files.

Reviewer #1: No

Reviewer #3: **Yes: **Ngozi Virginia Aikpokpo

Reviewer #4: No

Reviewer #5: No

---

## [Author Response · Author response to Decision Letter 1]

11 Jun 2024

Reviewer’s comments

Comment 1: The candidate uploaded the corrected manuscript with Track changes but not to the unmarked manuscript with no Track changes. Table 6 needs to be included in the corrected manuscript, and there was no corresponding correction in the abstract section. 

Response: We have provided the marked and clean copies. 

Comment 2: Hence, the corrected Manuscript in the unmarked should be uploaded, the table numbering corrected, and the abstract should reflect the changes made in the new tables 8.

Response: Thanks for the suggestions. We also made changes in the abstract.

Comment 3: Abstract: Page 2, Line 22-23 - Based on the changes the authors have effected in the body of the manuscript, the following statement should be corrected: “Risk factors for significant kidney damage were older age (OR=1.08, 95% CI 1.06-1.1, p<0.001)

Specifically, I request that you review the abstract to ensure it accurately reflects the changes made in the manuscript. This includes the correction of the statement regarding the risk factors for significant kidney damage.

Response: We have made changes as suggested.

Comment 4: Page 15, line 289: In line with reviewers' comments, the authors have changed the data points for Age on Table 8 (Multiple regression). Please make the same changes in the text and in the abstract. 

Response: We also made changes as suggested.

Comment 5: Similarly, the authors have removed urine protein from the table. Please make the same change in the abstract.

Table 8: Please clarify if the variable ‘Age’ as used in the logistic regression models was a continuous or categorical variable.

Response: Age was used as a continuous variable.

Comment 6: Limitations: Page 20, line 392—Being a facility-based study with respondents recruited from the outpatient clinic limits the ability to generalise the findings to the general population. Please kindly indicate this fact as part of the study limitations.

Response: We have made changes as suggested.

Comment 7: References: Please kindly ensure that all the references are in alignment with the journal requirements

Response: We have made changes as suggested.

---

## [Editor Report · Decision Letter 2]

20 Jun 2024

Kidney Damage and Associated Risk Factors in the Rural Eastern Cape, South Africa: A Cross-Sectional Study

PONE-D-23-27552R2

Dear Dr. Rosales Gonzalez,

We’re pleased to inform you that your manuscript has been judged scientifically suitable for publication and will be formally accepted for publication once it meets all outstanding technical requirements.

Kind regards,

Ibrahim Sebutu Bello, MBBS, MPH, MD, FMCGP

Academic Editor

PLOS ONE

Additional Editor Comments (optional):

All issues raised have been addressed
---

## [Editor Report · Acceptance letter]

26 Jul 2024

PONE-D-23-27552R2 

PLOS ONE

Dear Dr. Rosales Gonzalez, 

I'm pleased to inform you that your manuscript has been deemed suitable for publication in PLOS ONE. Congratulations! Your manuscript is now being handed over to our production team.

Kind regards, 

on behalf of

Dr. Ibrahim Sebutu Bello 

Academic Editor

PLOS ONE